# Revisiting the ownership effect in adults with and without autism

**Marchella Smith[1], David Williams[1], Sophie Lind[2], Heather J. Ferguson[1]***

**1** School of Psychology, Keynes College, University of Kent, Canterbury, United Kingdom, **2** Department of Psychology, City, University of London, Northampton Square, London, United Kingdom

* h.ferguson@kent.ac.uk

**Data Availability Statement:** The datasets supporting this article are available on the Open Science Framework (osf.io/y2mf9).

**Funding:** The authors received no specific funding for this work.

## Abstract

Self-owned items are better remembered than other-owned items; this ownership effect reflects privileged processing of self-related information. The size of this ownership effect has been shown to decrease in neurotypical adults as the number of autistic traits increases, and is reduced in autistic adults. However, emerging evidence has questioned the reliability of these findings. This paper aimed to replicate previous work using well-powered, pre-registered designs, and Bayesian analyses. Experiment 1 (N = 100) found a significant ownership effect in neurotypical adults; however, the size of this was unrelated to individual differences in autistic traits. Experiment 2 (N = 56) found an ownership effect in neurotypical but not autistic adults. The findings suggest that individual differences in autistic traits in the neurotypical population do not impact the ownership effect, but a clinical diagnosis of autism might. We discuss how these findings can be explained by differences in psychological self-awareness in autism.

## Introduction

### Revisiting the ownership effect in adults with and without autism

The Self is widely thought to affect memory processing in a variety of ways [1–3]. This influence has been demonstrated empirically by research into self-biases in performance on memory tests. It is widely observed among neurotypical people that memory for self-referential information is superior to memory for information relating to other people. This self-bias, or "self-reference effect", is thought to be driven by preferential processing of self-relevant information relative to other-relevant information [4]. One of the most widely studied effects is the so-called "ownership effect", which reflects superior recall or recognition memory for objects previously encoded as being (notionally) owned by the self than objects previously encoded as being (notionally) owned by someone else [5, 6]. This self-bias is thought to reflect distinct cognitive processing of items considered extensions of the self [7]. Given that memory is influenced by the self, it follows that any individual with a diminished self-concept should have diminished and/or atypical memory capacity/profile. Potentially, this is highly relevant for our understanding autism spectrum conditions (ASC). Note that we acknowledge the debates about terminology used to describe autism, and therefore in this paper use a mixture of

**Competing interests:** The authors have declared that no competing interests exist.

commonly endorsed terms to reflect the fact that "there is no single way of describing autism that is universally accepted and preferred" [8].

ASC is a neurodevelopmental condition diagnosed on the basis of social-communication impairments, and a restricted and repetitive repertoire of behaviour and interests [8]. At the psychological level, ASC is thought to be characterised by difficulties in aspects of both self-awareness and memory, and it has been suggested that these two atypicalities are linked [9]. If self-awareness is diminished among autistic people, then the self should not structure memory in the same way that it does for most neurotypical people, and so autistic people should show a reduced (or absent) ownership effect on traditional paradigms.

Indeed, in a seminal study, Grisdale et al. (2014) found that autistic adults showed a significantly diminished ownership effect relative to that shown by age- and IQ-matched neurotypical adults. Whereas neurotypical participants showed a highly significant 9% advantage in memory for items they owned over items owned by an experimenter, autistic participants only showed a 1% (non-significant) advantage in memory for items owned by the experimenter over items owned by themselves. The between-group difference in the size of the ownership effect was large and statistically significant, in keeping with the theory that self-referential processing has a reduced/absent effect on memory processing among autistic adults. Furthermore, Grisdale et al. (Experiment 1) adopted an individual differences approach and found that the size of the ownership effect was associated moderately and significantly with the number of autistic traits reported among a group of 40 neurotypical individuals (i.e., the more autistic traits, the smaller the ownership effect). Since ASC is seen as a continuous spectrum rather than categorical, autistic traits are thought to vary continuously in the population, thus it is possible to learn about ASC by studying individual differences in autistic traits in a neurotypical population (although further analysis in a case-control sample is also necessary to check the validity.

Taken together, Grisdale et al.'s results appear in keeping with the theory that self-awareness is diminished in ASC. One possible consequence of atypical psychological self-awareness in ASC is that it causes people to value self-owned items differently, and this contributes to the reduced ownership effect seen among autistic people [10]. Hartley and Fisher (2018) found that when children were randomly assigned one of three toys to own, neurotypical children showed a preference to retain their assigned toy (i.e. they rarely traded it for a different item, thus showing an ownership effect) but autistic children frequently traded for a different toy that they preferred. This suggests that unlike neurotypical children, autistic children do not over-value self-owned items and do not experience a mnemonic advantage for these self-owned items. Therefore, impaired psychological self-awareness in ASC could also lead to an atypical ability to bind novel information to the self, which might contribute to the memory difficulties experienced by many autistic people [9, 11] and reduce the efficiency with which they update self-knowledge. This would perpetuate impairments in psychological self-awareness.

However, more recent findings are somewhat more difficult to interpret. Firstly, studies with neurotypical adults have found no significant association between autisitc traits (10-item AQ; or 50-item AQ) and self-bias magnitude in memory [12, 13], perception, or attention [13–15], which suggest that self-bias is not affected by autistic tendencies in neurotypical people. Additionally, in the perceptual domain, research has found no difference in self-bias between neurotypical and autistic adults [14]. Furthermore, Gillespie-Smith et al. (2018) used the ownership task in a case-control design and reported a significantly *enhanced* ownership effect in autistic children compared to that observed in either neurotypical children matched for chronological age or neurotypical children matched for verbal ability. This is the exact opposite of the pattern found by Grisdale et al. (2014, Experiment 2). Gillespie-Smith et al.

subsequently performed a median split on the number of parent-reported autistic traits among participants in the ASC group. They then compared the size of the ownership effect in the group with "severe" ASC (i.e. above-the-median scores) and those with "mild" ASC (i.e. below-the-median scores), and found that only the sub-group with mild ASC showed a statistically significant ownership effect. In one respect, the finding that the severe ASC sub-sample did not show a significant ownership effect appears to replicate Grisdale et al.'s results. However, the severe ASC sub-sample only had 11 participants and thus had low power to detect a true ownership effect. In fact, the effect size for the contrast between self- and other-owned items in the severe ASC sample was large ($d = 0.78$) and borderline significant (p < .08). In other words, even the severe ASC sub-sample preferentially recognised self- over other-owned items to a statistically large extent. Indeed, the size of the ownership effect in the severe ASC sub-sample was actually *larger* than in either of the two neurotypical sub-samples (who showed only small ownership effects with associated effect sizes of $d = 0.45$ and $d = 0.47$, respectively) and not significantly smaller than that observed in the mild ASC sub-sample. Arguably, therefore, it is not appropriate to draw the conclusion that autistic participants in Gillespie-Smith et al.'s study showed anything other than a normal–or even enhanced–ownership effect, and this certainly contradicts the pattern observed by Grisdale et al. (2014).

An even more recent case-control study of the ownership effect in ASC was conducted by Wuyun et al. (2020). In their Experiment 1, autistic children were compared to young neurotypical children and age- and IQ-matched children with intellectual disability on a traditional ownership task. Children watched as an experimenter sorted pictures of items into a basket designated as the child's own, or a separate basket designated as the experimenter's own. Subsequent memory for items was assessed and the results indicated that both control groups showed a significant ownership effect, but the autistic group did not. This appears to replicate the findings of Grisdale et al. (2014), but arguably not those of Gillespie-Smith et al. (2018). In a second experiment, however, three new groups were tested on a version of the ownership paradigm that involved the child, rather than the experimenter, sorting items into their own and the other's baskets. Results showed that all three groups, including the ASC group, showed a significant ownership effect, contrary to findings from their Experiment 1 and the findings from Grisdale et al. (2014). The authors suggested that they expected these results on the basis that "the hands-on experience of identifying ownership would strengthen the connection between self and the self-related items". It is unclear why merely self-sorting would increase the salience of self-owned items, because participants sorted *both* self-owned and other-owned items and so motor traces associated with action should have scaffolded memory for both types of item [16–18]. More importantly, the self-sorting procedure used by Wuyun et al. in their Experiment 2 was similar to the procedure used by Grisdale et al. who observed an absence of the ownership effect in ASC, contrary to Wuyun et al.'s Experiment 2 results. Additionally, research has investigated the influence of the self on other cognitive domains, including perception and attention. In these domains research has often found intact self-bias in autistic adults [14].

The point in the analysis of previous research on the ownership effect in ASC is not to criticise the studies or suggest methodological flaws, but rather to highlight that findings are inconsistent across studies. Despite our own previous arguments that self-biases of various kinds are diminished in ASC, the inconsistency of results across studies has led us to be more cautious in drawing conclusions about the nature of self-awareness (and its relation to memory) in ASC. Therefore, the current study aimed to replicate the study by Grisdale et al. (2014), which produced the clearest evidence of a diminished ownership effect in ASC, but using well-powered sample sizes, pre-registered plans and more sensitive (Bayesian) statistical methods. We felt it important to try and replicate Grisdale et al. (2014) given the risk of drawing false

conclusions based on limited studies and the potential that this area of research might have been misled by publication biases [12]. Experiments 1 and 2 were fully pre-registered on the Open Science Framework (osf.io/y2mf9).

## Experiment 1 | Individual differences approach

### Method

**Participants.** A volunteer sample of 100 (16 male; 84 female) undergraduate students, aged 18–58 years ($M$ = 20.11; $SD$ = 4.80), from the University of Kent took part. An *a priori* power analysis using G*Power [19] revealed that 92 participants would be enough to detect a correlation between AQ [20] and level of ownership effect with correlation size $r$ = -0.33 ([6], Exp. 1) and power of .90. Grisdale et al. (2014) included 40 (2 male; 38 female) neurotypical undergraduate students aged 18–24 years, with a mean AQ score of 12.52 (SD = 5.94; range: 2–23). Thus, our mean AQ score is higher, and the range is larger than that in Grisdale et al (indicating greater autistic traits and greater variance in our sample).

All participants had normal or corrected-to-normal hearing and vision and had no history of psychiatric or neurological conditions (self-report). All participants provided written informed consent and following completion of the tasks were fully debriefed and received partial course credits for their participation in the current (and an additional unrelated) experiment. Ethical approval was provided by the School of Psychology's Research Ethics Committee at the University of Kent, and data collection took place in 2021. During and after data collection, no information that could identify individual participants was recorded.

**Ownership task.** *Materials and stimuli*. The stimuli used were identical to those used by Grisdale et al. (2014). This comprised of 222 laminated picture cards (855mm x 685mm) depicting colour photographs of items commonly sold in a supermarket (e.g., pepper, kettle, washing-up liquid; see Item 1 in Appendices). The 222 items were divided into three groups of 74 items. The mean word length and number of syllables of items did not significantly differ between the three groups [6]. For counterbalancing purposes, the condition allocated to each group of items was rotated across participants so that each group of items was used for self-owned, other-owned and "lure" item conditions (lure items were only shown in the recognition phase). In the study phase, participants were either presented with red-bordered cards depicting self-owned items and blue-bordered cards depicting other-owned items or vice-versa. Therefore, there were twelve versions of the task, of which participants were randomly allocated to one version. In the recognition test phase, all three sets of items (self-owned; other-owned; lure items) were presented on a card with no coloured border surrounding. Identical red and blue shopping baskets were used to place self-owned and other-owned items during the study phase.

*Procedure and scoring*. The procedure replicated that of Grisdale et al. (2014, Exp. 1). In the *study phase*, participants were instructed to sort a pile of 148 (74 red-bordered and 74 blue-bordered) laminated picture cards into their own or the experimenter's shopping basket according to who owned the item (see Fig 1). Participants were asked to attend to the items on each card, and sort them into the associated basket upside-down, one at time. They were told that, for the duration of the experiment, they "owned" the red basket and all of the items presented on a red-bordered card, whereas the experimenter "owned" the blue basket and all the items presented on a blue-bordered card (colour association was counterbalanced across participants). The cards were presented in a random order for each participant with the exception that no more than four cards with the same coloured border could appear in a row. In the subsequent unexpected *recognition test phase*, participants were presented with 222 picture cards, 148 of which depicted the items presented in the study phase and 74 of which were "lure"

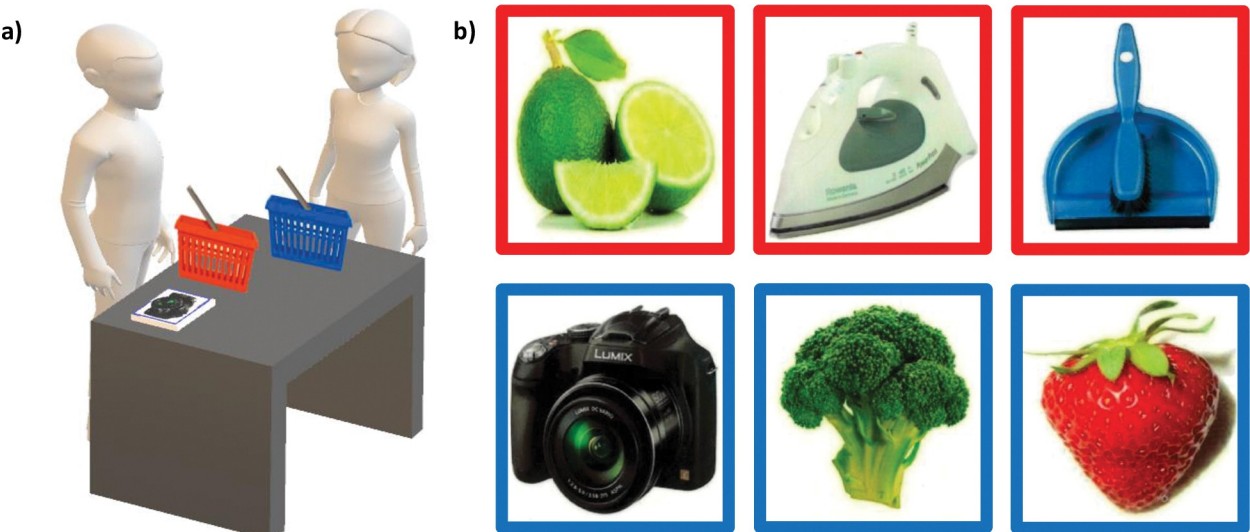

**Fig 1. Illustration of the ownership task set up (created in Microsoft Paint 3D).** (a) Experimenter/other (female with blue basket) and participant/self (male with red basket), and (b) example items owned by the self (red bordered cards) and owned by the other (blue bordered cards). The colour association was counterbalanced across participants (i.e. half the participants were allocated the red items, and half the participants were allocated the blue items).

items that had not previously been studied. None of the picture cards in the recognition test phase had a coloured border and participants were asked to answer whether they did or did not see each item at study (yes-no recognition test).

In line with Grisdale et al., each participant's hit rate (proportion of items correctly recognised as having been presented at study), false alarm rate (proportion of lure items that were incorrectly judged as having been presented at study), and corrected hit rate (hit rate *minus* false alarm rate) were calculated for each condition (self-owned/other-owned). Higher corrected hit rates reflected participant's better recognition memory. The size of the ownership effect was calculated as the corrected hit rate for self-owned items minus the corrected hit rate for other-owned items. The larger the resulting value, the greater the self-bias/ownership effect.

**Autism-spectrum Quotient.** Participants completed the Autism-spectrum Quotient (AQ; [20]). This is a standardized 50-item self-report questionnaire used to measure people's number of autistic traits. Participants were asked the extent to which they agree (definitely agree, slightly agree, slightly disagree, definitely disagree) with each statement about the self (e.g., "I find social situations easy"). The dependent variable is the total AQ score (ranging from 0–50), with higher scores indicating a greater number of autistic traits [21]. Good test-retest reliability scores have been observed (range between $r = .70$ and $r = .95$) [22]. In the current sample, the mean AQ score was 16.98 ($SD = 6.51$; ranging from 2–43).

**Statistical analysis.** As well as conducting standard inferential statistical analyses, we included Bayesian analysis to interpret our results (conducted using JASP 0.14.1; [22]) because Bayesian analysis enables a more graded interpretation of data compared to simply using $p$ values or effect sizes, by estimating the relative strength of the alternative hypothesis over the null hypothesis or vice versa (e.g., [23, 24]). Bayes factor ($BF^{10}$) $< 1$ provides evidence for the null hypothesis ($<0.33$ provides firm evidence), whereas Bayes factors $> 3$, $>10$, $> 30$, and $>100$ provides firm, strong, very strong, and decisive evidence for the alternative hypothesis respectively.

## Results

To investigate the effect of ownership on corrected hit rate, a repeated measures ANOVA was conducted with Referent (self-owned/other-owned) as the within-subjects independent variable and corrected hit rate as the dependent variable (see Table 1 for descriptive statistics). All assumptions for the current analysis were met, and alpha was set to .05. The ANOVA revealed a significant main effect of Referent, $F(1, 99) = 6.01$, $p = .02$, $\eta_p^2 = 0.06$, $BF^{10} = 2.31$, reflecting significantly greater corrected hit rate for self-owned compared to other-owned items. The data were also analysed categorically. If participants' corrected hit rate for self-owned items was higher than that of other-owned items, then they were considered to have shown an ownership effect. By this definition, 52/100 (52%) participants showed an ownership effect.

In addition to the main pre-registered analyses on corrected-hit rate, we ran analyses on d-prime ($d'$), which combines normalised hit rate and false alarms as an indicator of participants' sensitivity to detect signal among noise. When using d-prime ($d'$) as the dependent variable, a repeated measures ANOVA with Referent (self-owned/other-owned) as the within subjects independent variable showed a significant main effect of Referent, $F(1, 99) = 8.07$, $p = .01$, $\eta_p^2 = 0.08$, $BF_{10} = 5.76$, reflecting significantly greater $d'$ for self-owned ($M = 1.92$, $SD = .99$) compared to other-owned ($M = 1.84$, $SD = .99$) items.

A Pearson's correlation analysis examined the degree to which the size of the ownership effect was associated with individual differences in autistic traits (using AQ scores; see Fig 2). This revealed no significant correlation between the size of the ownership effect and total AQ score, $r = -.02$, $p = .81$, $BF^{10} = 0.13$. We also directly compared the correlation between size of the ownership effect and the AQ score from the current sample with that of Grisdale et al. (2014). A one-tailed Fisher Z test revealed that the correlation was significantly weaker in our sample than Grisdale et al.'s (2014; $r (40) = -.33$) sample, $Z = 1.67$, $p = .047$.

## Discussion

The aim of Experiment 1 was first to test whether the ownership effect and the association between the size of the ownership effect and self-reported autistic traits (AQ) observed by Grisdale et al. (2014, Exp. 1) could be replicated in a well-powered sample with pre-registered design and analysis. Secondly, like Grisdale et al. (2014), our Experiment 1 took an individual differences approach to test whether the size of the ownership effect was associated with the number of autistic traits. The use of a larger sample of participants allows us to determine the reliability of this task in influencing an ownership effect in neurotypical individuals, before using it to draw conclusions about an intact or reduced ownership effect in a smaller sample of autistic participants.

Experiment 1 found a significant ownership effect in neurotypical adults whereby significantly more self-owned items were successfully recognised than other-owned items. This replicates the patterns reported in multiple studies of the ownership effect among neurotypical

**Table 1. Descriptive statistics for proportion of hit rates, false alarm rates and corrected hit rates in each condition in Experiment 1.**

|  | Condition | Mean | SD |
|---|---|---|---|
| Hit Rate | Self-owned | .658 | .184 |
|  | Other-owned | .637 | .180 |
| False Alarm Rate | Lure items | .112 | .105 |
| Corrected Hit Rate | Self-owned | .547 | .214 |
|  | Other-owned | .525 | .213 |

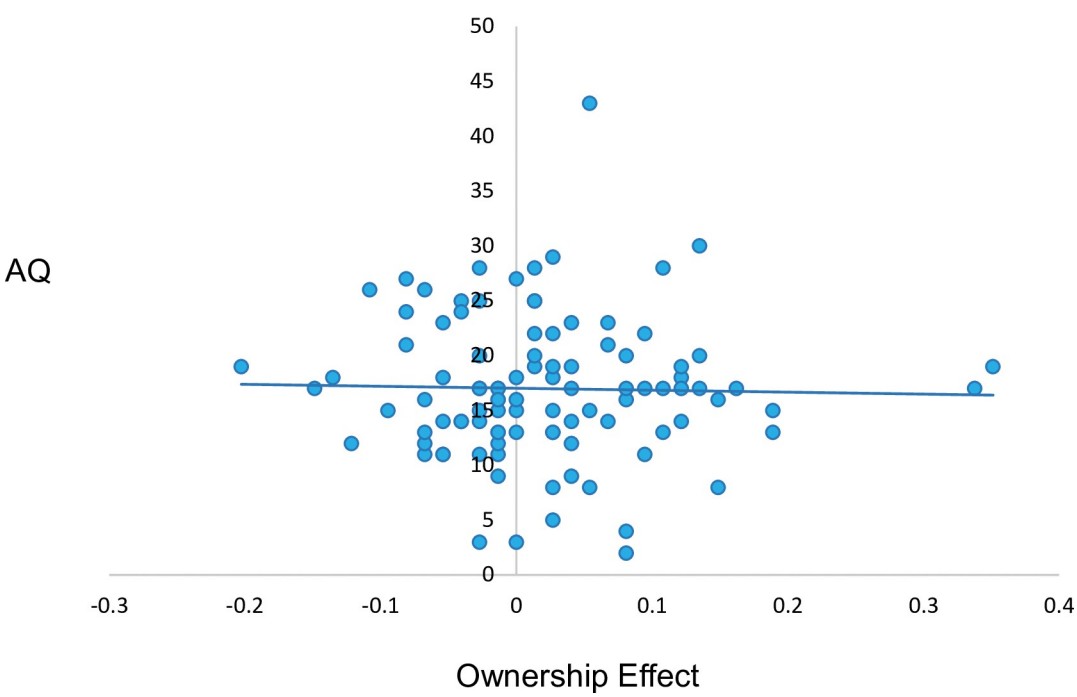

**Fig 2. Scatter plot depicting non significant correlation between size of ownership effect (corrected hit rate for self-owned items *minus* other-owned items) and AQ, in Experiment 1.**

individuals and supports the notion that items thought to be (in this case notionally) owned by the self, become a psychological extension of the self, which encourages enhanced cognitive processing [7]. It is important to note however, that the ownership effect size observed in the current study ($\eta_p^2 = 0.06$; 52% of participants displayed an ownership effect) is smaller than that observed by Grisdale et al. (2014, Experiment 1; $\eta_p^2 = 0.72$; 95% of neurotypical participants displayed an ownership effect). This difference is likely explained by subtle differences between studies in task instructions, the larger sample size in our study compared to Grisdale et al. (N = 100 *vs.* 40), and sample demographics.

Crucially, we did *not* observe a significant association between the size of the ownership effect and AQ. Bayesian analysis indicated that the null hypothesis was strongly supported by the data. Furthermore, the size of this correlation was significantly smaller than that of Grisdale et al. (2014). Hence this does not replicate Grisdale et al. (2014) and does not support our hypothesis. However, these results are in line with other emerging evidence that a) the ownership effect may not necessarily be diminished among autistic people [25, 26], and b) the size of other forms of memory self-bias are not significantly associated with the number of autistic traits manifest by neurotypical individuals, contrary to previous suggestions [12–14, 27]. Therefore, results from Experiment 1 appear to call into question the reliability of Grisdale et al.'s (2014, Exp. 1) findings and more generally, the claim that the ownership effect is significantly affected by autistic traits.

## Experiment 2 | Case/control approach

Whilst autistic traits vary within the neurotypical population, and hence an individual differences approach can inform us about the effects of autistic traits on the ownership effect, it does not rule out the possibility that individuals with clinical levels of autistic traits (i.e. an ASC diagnosis) produce outcomes that are qualitatively different compared to neurotypical

individuals with high autistic traits. To address this possibility, in Experiment 2 we used the same ownership task, but employed a case-control approach as Grisdale et al. (2014) did, to compare the size of the ownership effect between neurotypical and autistic adults.

## Method

**Participants.**   Participants were recruited from the Autism Research Kent participant database and included 28 neurotypical adults with no history of psychiatric or neurological conditions (self-report) and 28 adults with an autism spectrum disorder clinical diagnosis. Based on Grisdale et al. (2014), an *a priori* power analysis using G*Power revealed that 8 participants (4 per group) would be enough to detect a significant group × referent interaction with effect size $\eta_p^2 = 0.39$ and power of 0.90.

All participants had normal or corrected to normal hearing and vision. The two participant groups were matched on age, gender, and Intelligence Quotient (IQ, Wechsler Adult Intelligence Scale, WAIS-III or WAIS-IV; [28, 29]), but were not matched on Autism spectrum Quotient (AQ; 20). See Table 2 for participant demographics.

Participants in the ASC group received a clinical diagnosis of either autistic disorder (N = 6), autism spectrum disorder (N = 4) or Asperger's syndrome (N = 12) or not known (N = 6), of which official diagnostic information was checked. To assess the current autistic characteristics, 25/28 autistic participants were also assessed on module 4 of the Autism Diagnostic Observation Schedule (ADOS; [30]) by a trained, research-reliable researcher, and videos were double coded to ensure reliability of scoring (inter-rater reliability was found to be excellent with intraclass correlation of .89). In the autistic group, ADOS scores ranged from 1–21, with fifteen individuals scoring higher than 7 (i.e. the clinical cut off score). AQ scores ranged from 19–45 in the autistic group and from 3–28 in the neurotypical group.

Participants provided written informed consent and were fully debriefed following completion of the experiment. They received £10 per hour of their participation, plus additional travel expenses. Ethical approval was provided by University of Kent Research Ethics Committee, and data collection took place in 2022. During and after data collection, any information that could identify individual participants (i.e. contact details as part of the autism database) was kept separate from stored and analysed data to ensure anonymity of the data.

**Design.**   This was a 2 (Group: autistic/neurotypical) × 2 (Referent: self-owned/other-owned) mixed design with repeated measures on the second factor.

**Procedure and scoring.**   The stimuli, procedure and scoring were identical to those described for Experiment 1.

## Results

To investigate the effect of Group and Referent on corrected hit rate a 2 (Group: autistic/neurotypical) × 2 (Referent: self-owned/other-owned) mixed ANOVA was conducted, with

**Table 2. Comparison of participant characteristics between groups in Experiment 2.** IQ is based on 27 neurotypical and 28 autistic participants.

|  | Autistic (n = 28; 19 male) | Neurotypical (n = 28; 18 male) |  |  |  |
| --- | --- | --- | --- | --- | --- |
|  | Mean (SD) | Mean (SD) | *t* | *p* | *d* |
| Age (years) | 38.77 (12.69) | 39.47 (14.18) | 0.19 | .847 | 0.05 |
| IQ: Full Scale | 106.39 (16.27) | 110.59 (10.06) | 1.15 | .257 | 0.31 |
| IQ: Verbal | 104.61 (12.67) | 109.26 (10.97) | 1.45 | .152 | 0.39 |
| IQ: Performance | 108.36 (21.14) | 111.22 (11.16) | 0.63 | .535 | 0.17 |
| AQ | 30.79 (7.55) | 14.39 (6.36) | 8.60 | < .001 | 2.34 |
| ADOS: Total | 8.71 (5.23) | - | - | - | - |

**Table 3. Descriptive statistics for proportion of hit rates, false alarm rates and corrected hit rates in each condition in autistic and neurotypical participants (Experiment 2).**

|  | | Autistic | | neurotypical | |
| --- | --- | --- | --- | --- | --- |
|  | Condition | M | SD | M | SD |
| Hit Rate | Self-owned | .492 | .241 | .680 | .180 |
|  | Other-owned | .476 | .257 | .614 | .178 |
| False Alarm Rate | Lure items | .111 | .107 | .129 | .104 |
| Corrected Hit Rate | Self-owned | .380 | .269 | .551 | .182 |
|  | Other-owned | .365 | .276 | .485 | .191 |

repeated measures on the second factor (see Table 3 for descriptive statistics). This revealed a significant main effect of Referent, $F(1, 54) = 11.84$, $p = .001$, $\eta_p^2 = .18$, $BF^{10} = 19.23$, with a significantly greater corrected hit rate for self-owned ($M = .47$, $SD = .24$) compared to other-owned ($M = .43$, $SD = .24$) items. There was also a significant main effect of Group, $F(1, 54) = 5.62$, $p = .02$, $\eta_p^2 = .09$, $BF^{10} = 2.52$, showing that corrected hit rate was greater for the neurotypical ($M = .52$, $SD = .18$) than the autistic ($M = .37$, $SD = .27$) group.

Crucially, there was a significant Group × Referent interaction, $F(1, 54) = 4.54$, $p = .04$, $\eta_p^2 = .08$, $BF^{10} = 39.77$. Bonferroni corrected paired-samples t-tests revealed that neurotypical participants had a significantly greater corrected hit rate for self-owned compared to other-owned items, $t(27) = 3.84$, $p < .001$, $d = .73$, $BF^{10} = 47.78$. In contrast, autistic participants showed no significant difference in corrected hit rate according to referent, $t(27) = .95$, $p = .35$, $d = .18$, $BF^{10} = .30$. When splitting the autistic group according to those who did and did not score higher than 7 (i.e., the cut-off) on the ADOS, this revealed that neither the above cut-off, $t(14) = 1.83$, $p = .09$, $d = .47$ or below cut-off group, $t(8) = 1.07$, $p = .32$, $d = .36$, displayed a significant ownership effect. Furthermore, the corrected hit rate for self-owned items was significantly greater in the neurotypical than the autistic group, $t(54) = 2.78$, $p = .008$, $d = .74$, $BF^{10} = 5.94$; however, the corrected hit rate for other-owned items did not differ between groups, $t(54) = 1.90$, $p = .063$, $d = .51$, $BF^{10} = 1.80$.

When using $d'$ as the dependent variable instead of corrected-hit rate, a mixed ANOVA with Referent (self-owned/other-owned) as the within-subjects independent variable, and Group (autistic/neurotypical) as the between-subjects variable showed a significant main effect of Referent, $F(1, 51) = 13.88$, $p < .001$, $\eta_p^2 = 0.21$, $BF_{10} = 0.64$, reflecting significantly greater $d'$ for self-owned ($M = 1.53$, $SD = .83$) compared to other-owned ($M = 1.37$, $SD = .80$) items. There was no significant main effect of group, $F(1, 51) = 3.63$, $p = .06$, $\eta_p^2 = 0.07$, $BF_{10} = 0.02$, and no referent x group interaction, $F(1, 51) = 2.68$, $p = .11$, $\eta_p^2 = 0.05$, $BF_{10} = 0.63$.

The data were also analysed categorically, as in Experiment 1. This revealed that 20/28 (71.43%) neurotypical and 18/28 (65.29%) autistic participants showed an ownership effect. The between group difference was not statistically significant, $X^2 = 0.33$, $p = .57$.

**Association analyses.** An exploratory Pearson's correlation analysis replicated the analysis in Experiment 1 and Grisdale et al. to examine the degree to which the size of the ownership effect was associated with individual differences in autistic traits (using AQ scores; see Fig 3). This revealed no significant correlation between the size of the ownership effect and total AQ score in the neurotypical group, $r(26) = .12$, $p = .56$, $BF^{10} = 0.29$, or the autistic group, $r(28) = .03$, $p = .90$, $BF^{10} = 0.24$. There was also no significant correlation between the size of the ownership effect and total AQ score when both groups were combined into a larger sample, $r(54) = -.16$, $p = .24$, $BF^{10} = 0.34$.

We compared the correlation between size of the ownership effect and the AQ score from the current samples with that of Grisdale et al. (2014, Experiment 2). As in Experiment 1, a

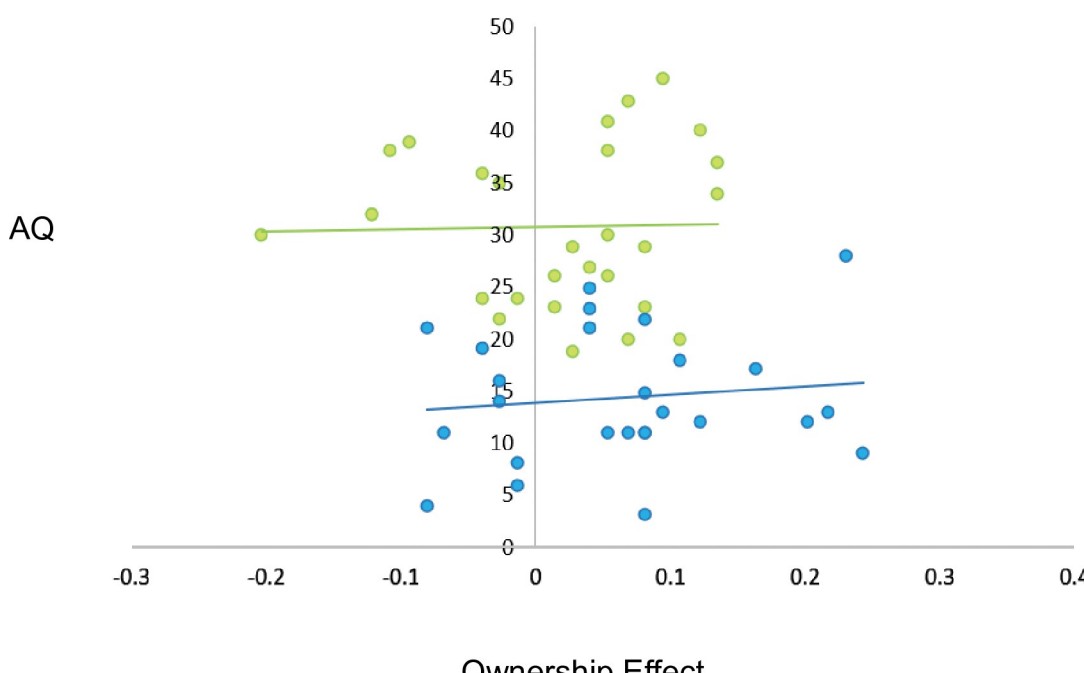

**Fig 3. Scatter plot depicting non-significant correlations between size of the ownership effect (corrected hit rate for Self *minus* Other) and AQ in neurotypical (blue) and autistic (green) groups, in Experiment 2.**

one-tailed Fisher Z test revealed that the correlation was significantly weaker in our neurotypical sample than Grisdale et al.'s (2014; $r(16) = -.11$) neurotypical sample, $Z = 0.67$, $p = .25$. The correlation was also significantly weaker in our autistic sample than Grisdale et al.'s (2014; $r(16) = -.10$) autistic sample, $Z = 0.38$, $p = .35$.

## Discussion

The significant interaction between group and referent revealed that unlike the neurotypical group, the autistic group did not demonstrate a statistically significant ownership effect; rather, the autistic group showed comparable memory for items encoded in relation to the self and in relation to the experimenter. Neurotypical participants were more accurate at recalling self-owned items than autistic participants ($p = .01$) but the two groups did not differ significantly in memory for other-owned items ($p = .063$). This suggests that autistic adults experienced greater difficulty with self-referential processing than other-referential processing. Nevertheless, the corrected hit rate for other-owned items was numerically higher among neurotypical participants than autistic participants, and at 1.80 the Bayes Factor of the between-group effect falls in the category of 'firm evidence for a difference'. The categorical analysis further revealed no between-group difference in the number of people who did/did not display memory biases in the direction of an ownership effect, which suggests that neurotypical participants were no more likely than autistic participants to display an ownership effect, and therefore the group difference reflects a reduced size of ownership effect in ASC rather than an absent one. Therefore, rather than there being a specific impairment in self-processing in autism, the pattern might reflect reduced memory accuracy more generally–i.e. involving both self- and other-processing.

The categorical percentages of ownership effect in both the neurotypical (71%) and autistic (65%) groups are higher than in the large neurotypical sample in Experiment 1 (52%).

Furthermore, the self-other difference in corrected hit rate suggests a larger ownership effect in the neurotypical group in Experiment 2 (0.07) compared to either the autistic group (0.02) or the neurotypical group (0.02) in Experiment 1. These differences might be influenced by differences in sample characteristics. In particular, the neurotypical sample in Experiment 1 was made up entirely of undergraduate students who completed the task for course credit, while participants in Experiment 2 were recruited from the Autism Research Kent participant database (made up of members of the local community who are interested in research) and were paid for their time. These different sample contexts might have influenced participants' attention/performance on the task due to differing levels of motivation [31, 32].

## General discussion

The current study aimed to investigate whether self-referential cognition is impaired in autistic adults relative to neurotypical adults. We tested the self-reference effect in memory–more specifically the ownership effect [8]–whereby neurotypical people typically display a memory advantage for items encoded in relation to the self (i.e., self-referentially) relative to those encoded in relation to other people [31, 32]). Experiment 1 took an individual differences approach by testing whether the ownership effect is associated with the number of autistic traits (measured by AQ; 20) possessed by neurotypical adults; Experiment 2 tested whether the ownership effect differs between neurotypical and autistic adults. We improved on previous studies that have examined this question (e.g., [6, 26, 33]) by using well-powered sample sizes, pre-registered plans and more sensitive (Bayesian) statistical methods.

Arguably, the key finding from the current experiments was that autistic participants in Experiment 2 showed a significantly reduced ownership effect compared to age- and IQ-matched neurotypical comparison participants. A significant group x referent interaction revealed that whereas neurotypical participants showed a moderate-to-large and statistically significant ownership effect in Experiment 2, autistic participants showed a small and statistically non-significant effect (i.e. comparable memory for items encoded in relation to the self and in relation to the experimenter). Moreover, whilst there was no significant group difference in memory for other-owned items, the neurotypical participants had higher memory for self-owned items than autistic participants. It is reassuring that these findings replicate closely those of Grisdale et al. (2012, Experiment 2) and add weight to the suggestion that this effect is reliably reduced among people with a clinical diagnosis of ASC. This is especially so, because the current study was a gold standard replication attempt, which was fully pre-registered, and included the same materials used by Grisdale et al. but with a sample that was almost 65% larger than the original investigation.

The ownership effect observed in the neurotypical group is in line with previous research that has used the ownership paradigm in neurotypical adults [5, 34] and children [34–36]. Among neurotypical people, memory for self-referent information is superior to memory for information relating to other people. This self-bias, or "self-reference effect", is thought to be driven by preferential processing of self-relevant information relative to other-relevant information [4, 37].

At the psychological level, ASC is thought to be characterised by difficulties in aspects of both self-awareness and memory, and it has been suggested that these two limitations are linked. If psychological self-awareness–and associated extended self-concept–is diminished among autistic people, then the self should not structure memory in the same way that it does in most neurotypical people. Consequently, we hypothesised that autistic people would show a reduced (or absent) ownership effect on traditional paradigms. The diminished ownership effect observed in the current study supports the proposal that psychological self-awareness is

diminished in ASC, and is consistent with research that has shown atypical self-processing in ASC across other domains (e.g. [38–40]). However, some caution is warranted before concluding that the reduced ownership effect observed among autistic participants in Experiment 2 reflects a reliable limitation in psychological self-awareness. While it is reassuring that this key finding replicated the original finding of Grisdale et al. (2012, Experiment 2), other results in the current study did not replicate the original results.

First, results from Experiment 1 challenged our pre-registered hypothesis. Although there was a clear ownership effect on memory, the size of this effect did not correlate with the number of autistic traits (this lack of correlation was replicated in our Experiment 2). Bayesian analysis also supported the null hypothesis, that no reliable association between the variables existed in the current sample. This suggests that the ownership effect is not affected by autistic tendencies in a neurotypical population (echoing findings by 25,31). This pattern contrasts with Grisdale et al.'s (Experiment 1) original finding, that showed a moderate and statistically significant association between the size of the ownership effect and AQ total score. The sample of neurotypical participants in our Experiment 1 was over twice the size of Grisdale et al.'s Experiment 1 sample, thus low statistical power is unlikely to explain the non-significant relation in the current study. Instead, it may be that the findings of an association between the ownership effect and number of ASC traits among non-autistic participants reported by Grisdale et al. is not reliable.

Second, in Experiment 2 Chi-squared analysis revealed that there was no categorical difference between groups in the number of people who did/did not display an ownership effect, which suggests that although the size of the ownership effect was smaller in ASC (as shown by our continuous variable analysis), autistic participants were no less likely than neurotypical participants to display an ownership effect. This pattern also contradicts Grisdale et al. (2014; Experiment 2), who found that the ownership effect was more likely in neurotypical (100% *vs*. 71% in the current study) than autistic (44% *vs*. 65% in the current study) participants. The sample size in Grisdale et al.'s (2014) Experiment 2 (16 participants per diagnostic group) was smaller than the current Experiment 2 (28 per diagnostic group), and this difference might account for the discrepancy.

If our current finding of a *non*-significant association between the ownership effect and ASC traits is reliable, then this raises questions on how this can be reconciled with the significantly reduced ownership effect among autistic participants in both Grisdale et al. (Experiment 2) and the current Experiment 2. There are at least two possible explanations for this discrepancy. First, it is possible that qualitative differences exist between autistic traits in the general population and autistic traits in those diagnosed with ASC. This is not the prevailing view in the field currently, which is why Grisdale et al. and the current authors took an individual differences approach in a first experiment. However, there is an emerging view that there may be differences in the underlying basis of autistic traits in diagnosed and undiagnosed individuals [40], and there are findings that suggest distinct phenotypes can be observed [e.g., 41]. It is therefore possible that no reliable association exists between autistic traits and the ownership effect in the general population, but that individuals with a clinical diagnosis of ASC nonetheless show a reliably diminished effect. Second, it is possible that the AQ is not a reliable or valid measure of autistic traits. Psychometric evaluations of the AQ have suggested that some factors existing in the scale are uncorrelated, or negatively correlated [42, 43]. If so, then total AQ scores may not be meaningfully interpreted [43], and so any correlation or lack of correlation with task performance similarly lacks validity. Some argue that considering the AQ subscales rather than total AQ might be a more reliable indicator of autistic traits [44]. To minimise the potential of these issues, a-priori power analyses based on previous research were conducted to predict the minimum sample size required to detect significant correlations

between autistic traits and task performance based on Grisdale et al.'s (2014) research that included similar samples and measures as ours.

One possible consequence of diminished psychological self-awareness in ASC may be that it causes people to value self-owned items differently, and this contributes to the reduced ownership effect seen among autistic adults. Hartley and Fisher (2018) found that when children were randomly assigned one of three toys to own, neurotypical children showed a preference to retain their assigned toy (i.e. they rarely traded it for a different item, thus showing an ownership effect) but autistic children frequently traded for a different toy that they preferred. This suggests that unlike neurotypical children, autistic children do not over-value self-owned items and do not experience a mnemonic advantage for these self-owned items. However, when children *self-selected* a toy to own, neurotypical and autistic children rarely traded it, showing that both groups over-valued their new toy relative to a toy owned by someone else, and experienced an ownership effect ([10]; Experiment 2). Importantly, while neurotypical children continued to show a preference for their original self-selected toy compared to a replica, autistic children showed no preference. Therefore, when both items have identical material qualities, mere ownership of one item does not influence the preferences of autistic children as it does in neurotypical children. This would explain the diminished ownership effect observed in subsequent memory, observed in Experiment 2, and in previous studies (e.g., Grisdale et al., 2014). Together, these findings suggest that autistic people elicit a more economically rational strategy that bases evaluation more on material qualities rather than ownership.

Another potential explanation for the reduced ownership effect in ASC relates to differences in their use of linguistic cues and their subsequent ability to identify ownership. Hartley et al. (2021) found that children with ASC were less accurate than neurotypical children (matched on receptive language) at tracking owner-object relationships *only* when relationships between objects and owners were described using *possessive pronouns* (e.g., "This pencil case is yours" and "Which pencil case is mine?"). When ownership relationships were described using *proper pronouns* (e.g., "This is Nina's lunchbox" and "Which lunch box is John's?"), the ability to track owner-object relationships was equivalent between groups [45]. *Proper pronouns* have a fixed referent (e.g., Nina is always the same person). This one-to-one mapping has minimal referential ambiguity and supports cross-situational associative learning. However, *possessive pronouns* can have different referents depending on who is speaking (e.g., "yours" can refer to an object that is owned by the self or someone else, depending on who says it). Tracking who the referent is therefore requires deictic shifting, whereby the speaker/listener must repeatedly remap the same pronoun to different referents depending on who is speaking [45, 46].

The current finding that self-referential cognition is diminished in autistic adults is a direct contrast to the patterns reported in *memory-related* studies among autistic children, where either no group difference or an enhanced effect in the autistic group was found ([25, 33; Experiment 2). It is hard to reconcile these differences in terms of developmental differences (i.e. that psychological self-awareness is intact in childhood but becomes impaired in adulthood), so it is likely that the discrepant findings across studies reflect differences in task format and sample characteristics. For example, in Gillespie-Smith et al.'s (2018) modified ownership task, dyads included pairs of participants who took it in turns to sort cards into the appropriate basket, rather than a participant and experimenter (as used in our studies, [6, 26]); therefore, the experimenter explains the instructions to two participants rather than one (as in the current study). In this case, using only possessive pronouns (as in the current study, e.g., experimenter refers to "your" items) would cause ambiguity as to which participant is being referred to. It is more likely that proper pronouns were used (e.g., "Half of the items are Jack's, and half

of the items are James') to refer to each participant more explicitly. If so, this would not require deictic shifting. Therefore, it might be that the intact ownership effect in children with ASC *with low social difficulties* observed by Gillespie-Smith et al. (2017) is supported by the reduced ownership ambiguity, which allowed more differential attention applied to own and other items. Additionally, different social contexts were established during the tasks (i.e. peer-to-peer *versus* more formal adult-to-child), and this may have influenced the success with which cognitive processes were deployed or managed. Furthermore, Gillespie-Smith et al. (2018) only found intact ownership in children with *low* social difficulties, and not those with *high* social difficulties, so it might be that this latter group is more comparable to the adult autistic samples used here and in Grisdale et al. (2014). Importantly, since there were only 11 participants in each of the low and high social difficulties subsamples, Gillespie-Smith et al.'s (2018) sub-group analysis is likely to be poorly powered to reliably detect effects. The results of Wuyun et al.'s (2020) Experiment 2 are harder to reconcile with the current study given that task instructions and set-up appeared to be identical. The key difference between studies was that Wuyun and colleagues reduced the number of items to encode (6 items allocated each to the self and other, *versus* 74 here) to reduce task demands in their child sample. Therefore, Wuyun et al.'s (2020) study might not have elicited the level of encoding that would reveal an impairment in ASC; self-bias might be intact in ASC when the cognitive demands of the task are lower [47]. Nevertheless, the results from the current study are in keeping with the growing body of evidence showing differences in autistic children/adults ownership-related cognition in *non-memory-related* tasks [11, 45, 48, 49].

An ownership effect is thought to be an index of psychological self-awareness; in other words, enhanced memory for items encoded in relation to the self vs. someone else reflects greater *awareness* of our own, relative to other people's characteristics [5, 50]. However, an ownership effect might not be *dependent* on a second-order representation of self. The objects that are processed in relation to the self vs. other are not necessarily associated with the person that it is being encoded in relation to, and there is likely variance in the extent to which participants believe that items associated with the self or other are truly owned by that person. As such, the self-bias is driven by automatic up-regulation of attention to *anything* that participants are told to consider in relation to the self [51]. Therefore, this task might be measuring the *initial binding* of information to the self, but not tapping into the psychological effects associated with ownership, so might reflect something qualitatively different than an ownership effect that is developed organically when items are *genuinely* owned by the self.

Finally, it is important to consider the heterogeneity in ASC [40, 52]. It is possible that the discrepancy between studies is caused by differences in sample characteristics. For example, depression is found to influence memory [53] and depressive symptoms are highly co-morbid with ASC. Therefore, it is possible that differences in depression between samples might explain the difference in ownership between children (e.g., [25]) and adults (e.g., the current Experiment 2; 6). it would be useful for future research to control for depressive symptoms to rule out the possibility that this has confounded the results. Furthermore, future research could investigate the association between sub-scales of AQ and the ownership effect; more specific aspects of AQ might be more reliable predictors of the ownership effect than total AQ.

## Conclusion

In this paper, we have presented two pre-registered experiments that examined whether self-referential cognition is impaired in autistic adults. In Experiment 1, the ownership effect was replicated in a large neurotypical sample, but the size of this ownership effect was not associated with autistic traits. This suggests that individual differences in autistic traits do not impact

the ownership effect, but does not rule out the possibility that individuals with clinical levels of autistic traits (i.e. an ASC diagnosis) produce outcomes that are qualitatively different compared to neurotypical individuals with high autistic traits. Experiment 2 used a case-control design and replicated Grisdale et al.'s (2014; Experiment 2) key findings, showing that the ownership effect was significantly smaller (but just as likely to occur) in autistic adults. This suggests that psychological self-awareness is diminished in ASC, which diminishes integration of items into one's extended-self. Autistic people may therefore elicit a more economically rational strategy [11] that emphasises material qualities rather than ownership. The current study challenges previous studies that have found an intact or even enhanced ownership effect in autistic children [25, 26]. This discrepancy is hard to reconcile in terms of developing cognitive processes, and is likely to reflect methodological and demographic (e.g., social ability) differences between studies.

## Acknowledgments

Declarations of interest: none. The datasets supporting this article are available on the Open Science Framework (osf.io/n5mr3)

## Author Contributions

**Conceptualization:** Marchella Smith, David Williams, Sophie Lind.

**Data curation:** Marchella Smith.

**Formal analysis:** Marchella Smith, Heather J. Ferguson.

**Supervision:** Heather J. Ferguson.

**Writing – original draft:** Marchella Smith, David Williams.

**Writing – review & editing:** Sophie Lind, Heather J. Ferguson.

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
