## [Decision Letter · Decision Letter 0]

22 May 2023

PONE-D-23-08426Revisiting the Ownership Effect in Adults with and without AutismPLOS ONE

Dear Dr. Ferguson,

Thank you for submitting your manuscript to PLOS ONE. After careful consideration, we feel that it has merit but does not fully meet PLOS ONE’s publication criteria as it currently stands. Therefore, we invite you to submit a revised version of the manuscript that addresses the points raised during the review process.

I secured reviews from two experts in the field, both of whom value your contribution. Reviewer 1 explicitly lauds your commitment to Open Science practice, and I second this. Both reviewers agree that your introduction should provide a broader context for your own work. Both reviewers also raise excellent points about the interpretation of your results. Please address these points in your revision.

We look forward to receiving your revised manuscript.

Kind regards,

Johannes Hönekopp

Academic Editor

PLOS ONE

Journal Requirements:

Reviewers' comments:

Reviewer's Responses to Questions

**Comments to the Author**

1. Is the manuscript technically sound, and do the data support the conclusions?

Reviewer #1: Yes

Reviewer #2: Yes

2. Has the statistical analysis been performed appropriately and rigorously? 

Reviewer #1: Yes

Reviewer #2: Yes

3. Have the authors made all data underlying the findings in their manuscript fully available?

Reviewer #1: Yes

Reviewer #2: Yes

4. Is the manuscript presented in an intelligible fashion and written in standard English?

Reviewer #1: Yes

Reviewer #2: Yes

5. Review Comments to the Author

Reviewer #1: I appreciated the opportunity to review this interesting and well-written manuscript. The paper reports two experiments investigating how associating items with the self (vs. others) influences memory (i.e. an ownership effect). In Experiment 1, a large sample of neurotypical adults recognised items associated with the self significantly more accurately than items associated with others. The size of the ownership effect was not predicted by variability in autistic traits. Experiment 2 revealed differences between samples of autistic and neurotypical adults, notably that the ownership effect was larger for neurotypical adults. The findings are interpreted in terms of differences in psychological self-awareness that can characterise autism.

The experiments are methodologically rigorous and the samples are justified by a priori power analyses. The authors should be commended for diligently upholding Open Science principles. This research represents excellent practice.

In my view, the key areas for improvement concern the situation of the memory task and findings within the broader literature investigating ownership effects and their interpretation/explanation of results. I provide detailed comments/suggestions for each section below.

Abstract

* Very clear and well-written.

Introduction

* Page 5, paragraph 2: Typo: “Firstly, in neurotypical adult studies have found…” – omit “in”?

* The introduction provides detailed discussion of studies investigating the ownership effect in memory tasks. This is understandable given the authors’ task. However, the ownership effect has been investigated in other ways that would be worth highlighting – particularly as the strength of the effect may differ across methodologies. For example, in Hartley and Fisher (2018), autistic children clearly did not display a mere ownership effect when valuing/trading toys they were actually given to own and keep. It would be interesting to consider here – and perhaps revisit in the General Discussion in light of the results – how/why the ownership effect might differ across samples of autistic children/adults when tested on a memory task versus object preference/valuation tasks.

* There is a recent and relevant study that the authors may (or may not) wish to mention in their introduction: Hartley & Leeming (2022, Cog Dev) Could sensitivity to object authenticity be developmentally delayed in autism? Here, autistic adults were asked to value a range of hypothetical objects, including items belonging to the self. For the most part, neurotypical and autistic adults did not differ in how they valued items associated with self. The groups differed more in terms of how they valued objects belonging to other people.

Experiment 1

Methods

* Clearly reported and robust.

* In Experiment 1, the sample’s mean AQ score is 16.98. It would be helpful to indicate how this compares with the sample in Grisdale et al. (2014) that comparisons are being drawn with. Did the samples differ in their distribution of AQ scores and could this potentially explain differences in findings?

Results/Discussion

* I was somewhat surprised to see that only 52% of participants showed an ownership effect. How does this compare with rates in other studies employing similar methodologies? Notably, this proportion seems much lower than what we see in other tasks assessing ownership effects (e.g. studies involving ownership of real objects). In this Discussion, or perhaps the General Discussion, I think it would be worth reflecting on this point and considering to what extent this kind of memory task actually taps into psychological effects associated with ownership.

Experiment 2

Methods

* Clearly reported and robust.

Results/Discussion

* Analyses are reported very clearly.

* The ownership effect categorical classification percentages for the neurotypical (71.43%) and autistic participants (65.29%) in Experiment 2 are quite a bit higher than that of the much larger sample of neurotypical participants in Experiment 1 (52%). Why might this be?

General Discussion

* The comparisons between correct hit rates across samples and conditions are interesting. In Experiment 1, the difference between corrected hit rates for self and other items in a sample of 100 neurotypical adults was 0.022 (0.547 vs. 0.525). In Experiment 2, the difference between corrected hit rates for self and other items in a sample of autistic adults was 0.015 (0.38 vs. 0.365), just 0.007 less. Thus, the effect of Referent seems pretty similar between these two groups across Experiments. The difference between corrected hit rates for self and other items in the neurotypical sample in Experiment 2 was 0.066 (0.551 vs 0.485), which is larger than the difference for the autistic group but also larger than the difference for the neurotypical group in Experiment 1. These comparisons between groups require unpacking. Why is the Referent effect larger for neurotypical participants in Experiment 2 relative to Experiment 1, and why is it similar for autistic and neurotypical participants across studies?

* It strikes me that the biggest, and perhaps most meaningful, difference between autistic and neurotypical groups is in overall corrected hit accuracy – the corrected hit rates for both self and other items in the autistic sample are substantially lower than those observed in the two neurotypical samples. This point is not discussed at present, but warrants detailed theoretical consideration.

* Page 22, paragraph 2: I couldn’t really understand the authors’ point “Second, experimental designs are developed to, and hence become popular if, they produce robust/replicable effects – i.e., effects that are low in between-subject variance. However, low variance between-subjects makes these tasks unreliable/less effective for detecting individual differences (i.e. a correlational approach) which are dependent on this between-subject variance (Hedge et al., 2018). Therefore, it is also possible that some or all the tasks in this thesis were unsuitable for an individual differences approach.” The memory task used in these experiments surely generated sufficiently high variance between participants, making it ideal for examining individual differences? The reported data certainly don’t seem to be characterised by floor or ceiling effects.

* Building on paragraph 2 on page 23, it would be beneficial to reflect on how and why ownership effects may differ across tasks and the implications for studying autistic and neurotypical children/adults. How do the memory findings reported in this study fit in with the broader literature examining ownership effects in autism? Overall, how might the influence of ownership on cognition differ for autistic people?

* Page 24, paragraph 2: The theoretical explanation in this paragraph requires more detail. Why might differences in the ownership effect in autism be explained by differences in understanding ownership rights? The ownership effect measured in this study is a memory effect, whereas awareness of ownership rights is underpinned by sensitivity to cultural rules. Also, how might impairments in Theory of Mind increase the difficulty of mentally representing and tracking relationships between people and objects? I can see how differences in mentalistic reasoning might impact on understanding of feelings/emotions towards objects held by the self or others, but couldn’t tracking of people-object relationships be subserved by basic statistical learning which is typically intact in autism? That being said, a recent study by Hartley and Leeming (2022, JADD) suggests that autistic children may have difficulty tracking these invisible relationships when objects look similar, and don’t identify their objects with increased accuracy like neurotypical children do. A stronger link could also be drawn between understanding of possessive pronouns and identification of relationships between people and property (see Hartley, Harrison & Shaw, 2021, JADD).

* Page 24, paragraph 3: Given the similarity between corrected hit rates for neurotypical participants in Experiment 1 and autistic participants in Experiment 2, I’m not sure it is entirely accurate to say that these findings indicate that “self-referential cognition is impaired in autistic adults”. As noted above, the most prominent difference in the autistic group appears to be generally less accurate memory in the task for items associated with both the self and others.

Reviewer #2: I have read this paper about the ownership effect in autism with interest. Generally, meaningful, well-powered, preregistered replication attempts are admirable, and this one was carried out and described thoroughly. I do have a few comments, as I will outline below.

As a first general comment, I think there could be a somewhat more elaborate discussion of the current state of research on the self in autism (not just in relation to autistic traits), and how these results fit into that bigger picture, as the ownership paradigm is of course only one of several existing self-bias measures.

In a footnote, it is mentioned that the referent x group interaction was not present when using d’. I think discussing this would be worthwhile, also because, given the large percentage of autistic participants showing an ownership effect, the absence of a significant difference in corrected hit rate in that group seems surprising. From the scatterplot, there seems to be one autistic individual showing a particularly strong ‘opposite’ ownership effect, could this have influenced results?

In the Discussion, there is some redundant text (repetition) on page 20 (start of the first and second paragraph), I think this could be shortened.

As an explanation for the discrepancy between findings of Exp 1 and Exp 2, the authors mention that the task may be less reliable/effective in detecting individual differences. However, the data show otherwise, with only 52% actually showing the effect and quite large variation. It would be good to mention this, as it makes this explanation for this specific task less likely.

Minor comments, in order:

Introduction, page 3: I understand the choice of ASC in the light of the terminology discussion, but when discussing the DSM diagnosis, it would be good to mention that in the DSM, the official term is ASD?

Introduction, page 4: ‘smaller the ownership effect’: the word ‘the’ is missing. For the sentence after (about autism as a spectrum and variation in the population), it would be good to add a reference.

Methods, page 10: a sentence is presented twice here.

Methods, page 11: What ‘stretched beta prior width’ was chosen for the Bayesian analyses for correlations?

Intermediate discussion, page 13: it replicates the patterns reported in previous studies, but were effect sizes also similar?

Discussion, page 26: Do the authors have subscale data for the AQ scores in this study? In this case, could the proposed analysis (investigating the association of subscales with the ownership effect) not be done as an exploratory analysis on these data?

6. PLOS authors have the option to publish the peer review history of their article (what does this mean?). If published, this will include your full peer review and any attached files.

Reviewer #1: **Yes: **Calum Hartley

Reviewer #2: No

---

## [Author Response · Author response to Decision Letter 0]

24 Jul 2023

Dear Dr Johannes Hönekopp

Please find attached a copy of the revised manuscript, “Revisiting the ownership effect in adults with and without autism”. This is a revision of manuscript number PONE-D-23-08426, as invited by yourself. 

We are very grateful to you and the two expert reviewers of the previous submission for your helpful comments. These have given us excellent guidance on how to improve the paper, and we feel that it is much improved because of it. Below we provide a detailed summary of how we have addressed the individual comments raised by each reviewer and yourself. We have also revised the formatting of the manuscript and referencing style throughout, in line with PLOS ONE style requirements. All changed text is highlighted in yellow in the revised manuscript. 

Reviewer #1:

Page 5, paragraph 2: Typo: “Firstly, in neurotypical adult studies have found…” – omit “in”?

We have corrected this typo.

The introduction provides detailed discussion of studies investigating the ownership effect in memory tasks. This is understandable given the authors’ task. However, the ownership effect has been investigated in other ways that would be worth highlighting – particularly as the strength of the effect may differ across methodologies. For example, in Hartley and Fisher (2018), autistic children clearly did not display a mere ownership effect when valuing/trading toys they were actually given to own and keep. It would be interesting to consider here – and perhaps revisit in the General Discussion in light of the results – how/why the ownership effect might differ across samples of autistic children/adults when tested on a memory task versus object preference/valuation tasks

Thank you for highlighting this. We have extended our discussion of the boundary conditions and potential mechanisms of the ownership effect in the Introduction and Discussion.

There is a recent and relevant study that the authors may (or may not) wish to mention in their introduction: Hartley & Leeming (2022, Cog Dev) Could sensitivity to object authenticity be developmentally delayed in autism? Here, autistic adults were asked to value a range of hypothetical objects, including items belonging to the self. For the most part, neurotypical and autistic adults did not differ in how they valued items associated with self. The groups differed more in terms of how they valued objects belonging to other people.

Thank you for suggesting this research; we have added this citation to the discussion. 

In Experiment 1, the sample’s mean AQ score is 16.98. It would be helpful to indicate how this compares with the sample in Grisdale et al. (2014) that comparisons are being drawn with. Did the samples differ in their distribution of AQ scores and could this potentially explain differences in findings?

Thank you for raising this. We have added a footnote to Methods section, noting that, “Grisdale et al. (2014) included 40 (2 male; 38 female) neurotypical undergraduate students aged 18-24 years, with a mean AQ score of 12.52 (SD = 5.94; range: 2-23). Thus, our mean AQ score is higher and the range is larger than that in Grisdale et al (indicating greater autistic traits and greater variance in our sample)”.

I was somewhat surprised to see that only 52% of participants showed an ownership effect. How does this compare with rates in other studies employing similar methodologies? Notably, this proportion seems much lower than what we see in other tasks assessing ownership effects (e.g. studies involving ownership of real objects). 

We have added a note regarding the difference in rates between studies: “…the ownership effect size observed in the current study (ηp2 = 0.06; 52% of participants displayed an ownership effect) is smaller than that observed by Grisdale et al. (2014) (ηp2 = 0.72; 95% of participants displayed an ownership effect)”. 

In this Discussion, or perhaps the General Discussion, I think it would be worth reflecting on this point and considering to what extent this kind of memory task actually taps into psychological effects associated with ownership. 

We have added a paragraph at the end of the general Discussion to relate the findings to psychological mechanisms.

The ownership effect categorical classification percentages for the neurotypical (71.43%) and autistic participants (65.29%) in Experiment 2 are quite a bit higher than that of the much larger sample of neurotypical participants in Experiment 1 (52%). Why might this be? The comparisons between correct hit rates across samples and conditions are interesting. In Experiment 1, the difference between corrected hit rates for self and other items in a sample of 100 neurotypical adults was 0.022 (0.547 vs. 0.525). In Experiment 2, the difference between corrected hit rates for self and other items in a sample of autistic adults was 0.015 (0.38 vs. 0.365), just 0.007 less. Thus, the effect of Referent seems pretty similar between these two groups across Experiments. The difference between corrected hit rates for self and other items in the neurotypical sample in Experiment 2 was 0.066 (0.551 vs 0.485), which is larger than the difference for the autistic group but also larger than the difference for the neurotypical group in Experiment 1. These comparisons between groups require unpacking. Why is the Referent effect larger for neurotypical participants in Experiment 2 relative to Experiment 1, and why is it similar for autistic and neurotypical participants across studies?

Thank you for raising this. We now highlight this point in the Discussion for Experiment 2, and note that differences in sample characteristics (including student sample in Experiment 1 and a higher overall group mean AQ) is likely to explain differences in classification %.

It strikes me that the biggest, and perhaps most meaningful, difference between autistic and neurotypical groups is in overall corrected hit accuracy – the corrected hit rates for both self and other items in the autistic sample are substantially lower than those observed in the two neurotypical samples. This point is not discussed at present, but warrants detailed theoretical consideration. 

We have extended the discussion for Experiment 2 to highlight that when the main effect of group was broken down to look at the group x referent interaction, there is no significant between group difference in memory accuracy for other owned items, but there is for self-owned items. This suggests that the impairment is in self-processing but not other processing.

Page 22, paragraph 2: I couldn’t really understand the authors’ point “Second, experimental designs are developed to, and hence become popular if, they produce robust/replicable effects – i.e., effects that are low in between-subject variance. However, low variance between-subjects makes these tasks unreliable/less effective for detecting individual differences (i.e. a correlational approach) which are dependent on this between-subject variance (Hedge et al., 2018). Therefore, it is also possible that some or all the tasks in this thesis were unsuitable for an individual differences approach.” The memory task used in these experiments surely generated sufficiently high variance between participants, making it ideal for examining individual differences? The reported data certainly don’t seem to be characterised by floor or ceiling effects. 

Thank you for pointing this out. We agree that this point was unclear and have removed it from the manuscript.

Building on paragraph 2 on page 23, it would be beneficial to reflect on how and why ownership effects may differ across tasks and the implications for studying autistic and neurotypical children/adults. How do the memory findings reported in this study fit in with the broader literature examining ownership effects in autism? Overall, how might the influence of ownership on cognition differ for autistic people? 

We have added this additional reflection on effects in autism to the General Discussion, and added discussion linking these effects to language and ToM in autism.

Page 24, paragraph 2: The theoretical explanation in this paragraph requires more detail. Why might differences in the ownership effect in autism be explained by differences in understanding ownership rights? The ownership effect measured in this study is a memory effect, whereas awareness of ownership rights is underpinned by sensitivity to cultural rules. Also, how might impairments in Theory of Mind increase the difficulty of mentally representing and tracking relationships between people and objects? I can see how differences in mentalistic reasoning might impact on understanding of feelings/emotions towards objects held by the self or others, but couldn’t tracking of people-object relationships be subserved by basic statistical learning which is typically intact in autism? That being said, a recent study by Hartley and Leeming (2022, JADD) suggests that autistic children may have difficulty tracking these invisible relationships when objects look similar, and don’t identify their objects with increased accuracy like neurotypical children do. A stronger link could also be drawn between understanding of possessive pronouns and identification of relationships between people and property (see Hartley, Harrison & Shaw, 2021, JADD). – 

Thank you for pointing out this gap. We have significantly expanded the Discussion to discuss the findings in relation to theories/models of ToM and language, and have included a more direct explanation of how this might have differed between studies. 

Page 24, paragraph 3: Given the similarity between corrected hit rates for neurotypical participants in Experiment 1 and autistic participants in Experiment 2, I’m not sure it is entirely accurate to say that these findings indicate that “self-referential cognition is impaired in autistic adults”. As noted above, the most prominent difference in the autistic group appears to be generally less accurate memory in the task for items associated with both the self and others. 

We have replaced ‘impaired’ with “diminished” throughout the manuscript. While we agree that the main effect of group in Experiment 2, showing that corrected hit rate was greater for the neurotypical than the autistic, the key finding is that this effect was qualified by a group x referent interaction, meaning that this reduced accuracy in ASC is only significant in the self-owned trials and not in the other-owned trials (so not an overall reduction in memory accuracy in ASC).

Reviewer #2: 

As a first general comment, I think there could be a somewhat more elaborate discussion of the current state of research on the self in autism (not just in relation to autistic traits), and how these results fit into that bigger picture, as the ownership paradigm is of course only one of several existing self-bias measures. 

We have included additional studies using case control designs, including where no between group difference was found in self-biased perception (Williams et al. 2018), and highlight where further case-control studies are needed. 

In a footnote, it is mentioned that the referent x group interaction was not present when using d’. I think discussing this would be worthwhile, also because, given the large percentage of autistic participants showing an ownership effect, the absence of a significant difference in corrected hit rate in that group seems surprising.

Thank you for highlighting this. We agree that this non-significant main effect of group and interaction on d’ is surprising given the significant effects on these terms in the main ANOVA of corrected hit rate. This pattern suggests that the two diagnosis groups may differ in their response biases to self- and other-owned items in this task, but not in their sensitivity (i.e. the two groups are comparable in discriminating the signal for self-owned items better than for other-owned items). While we note this effect in a footnote for transparency, we focus our discussion on the results from corrected hit rates because this is the measure used in previous studies.

From the scatterplot, there seems to be one autistic individual showing a particularly strong ‘opposite’ ownership effect, could this have influenced results? 

Thank you for highlighting this data point. We have re-run the analysis without this outlier and the results are the same, thus this outlier is not having a significant impact on the results and we have retained in the full sample. 

In the Discussion, there is some redundant text (repetition) on page 20 (start of the first and second paragraph), I think this could be shortened. 

We have revised this section to remove redundancies.

As an explanation for the discrepancy between findings of Exp 1 and Exp 2, the authors mention that the task may be less reliable/effective in detecting individual differences. However, the data show otherwise, with only 52% actually showing the effect and quite large variation. It would be good to mention this, as it makes this explanation for this specific task less likely. 

We have revised this section to remove the discrepancy.

Minor comments:

Introduction, page 3: I understand the choice of ASC in the light of the terminology discussion, but when discussing the DSM diagnosis, it would be good to mention that in the DSM, the official term is ASD? 

We have replaced ASC with autism spectrum disorder in the participant section when outlining the DSM diagnosis inclusion criteria.

Introduction, page 4: ‘smaller the ownership effect’: the word ‘the’ is missing.

We have corrected this typo.

For the sentence after (about autism as a spectrum and variation in the population), it would be good to add a reference. 

We have added relevant references.

Methods, page 10: a sentence is presented twice here. 

We have removed this repeated sentence. 

Methods, page 11: What ‘stretched beta prior width’ was chosen for the Bayesian analyses for correlations? 

All Bayes factors compare to the null rather than compare to the best model. 

Intermediate discussion, page 13: it replicates the patterns reported in previous studies, but were effect sizes also similar? 

We have clarified in the manuscript (Experiment 1) that, “It is important to note however, that the ownership effect size observed in the current study (ηp2 = 0.06; 52% of participants displayed an ownership effect) is smaller than that observed by Grisdale et al. (2014) (ηp2 = 0.72; 95% of participants displayed an ownership effect).”, and for Experiment 2 that “It is important to note that the Group x Referent effect observed in the current study (ηp2 = .08) is smaller than that observed by Grisdale et al. (2014) (ηp2 = .39).” 

Discussion, page 26: Do the authors have subscale data for the AQ scores in this study? In this case, could the proposed analysis (investigating the association of subscales with the ownership effect) not be done as an exploratory analysis on these data?

We do not have subscale data for the full set of participants.

---

## [Decision Letter · Decision Letter 1]

3 Sep 2023

PONE-D-23-08426R1Revisiting the Ownership Effect in Adults with and without AutismPLOS ONE

Dear Dr. Ferguson,

Thank you for submitting your manuscript to PLOS ONE. Your reviewer and I both think your manuscript importantly improved. However, the reviewer still raised a number of meaningful points that I would like you to address in another revision.

We look forward to receiving your revised manuscript.

Kind regards,

Johannes Hönekopp

Academic Editor

PLOS ONE

Reviewers' comments:

Reviewer's Responses to Questions

**Comments to the Author**

1. If the authors have adequately addressed your comments raised in a previous round of review and you feel that this manuscript is now acceptable for publication, you may indicate that here to bypass the “Comments to the Author” section, enter your conflict of interest statement in the “Confidential to Editor” section, and submit your "Accept" recommendation.

Reviewer #1: (No Response)

2. Is the manuscript technically sound, and do the data support the conclusions?

Reviewer #1: Partly

3. Has the statistical analysis been performed appropriately and rigorously? 

Reviewer #1: Yes

4. Have the authors made all data underlying the findings in their manuscript fully available?

Reviewer #1: Yes

5. Is the manuscript presented in an intelligible fashion and written in standard English?

Reviewer #1: Yes

6. Review Comments to the Author

Reviewer #1: I appreciated the opportunity to review the revised version of this interesting manuscript. The authors have made a commendable effort to address the first reviews. However, in my view, some issues that were present in the initial submission remain (particularly concerning theoretical interpretation of results) and some of the newly added material requires more careful integration and/or explanation.

Introduction

* Page 4 paragraph 2: When describing Grisdale et al. (2014) it is stated that neurotypical participants showed a 9% advantage in memory for self-owned items over other-owned items. When describing the results of the autistic participants, it would be helpful to also include a % for accurate comparison.

* Page 4 paragraph 2: New text at the end of para 2 seems to be presented in a smaller font.

* Page 5 paragraph 2: Possible typo “Firstly, neurotypical adults studies…” perhaps reword to “Firstly, studies of neurotypical adults”?

* Page 5 paragraph 2: With the added text, sentence 2 is extremely long. Recommend breaking up by starting a new sentence from “Additionally, in the perceptual domain…”

Experiment 1

Results

* Page 11, footnote: For readers who are unfamiliar with d-prime, it might be worth providing a brief definition and explanation for its relevance to this study.

* Page 12 paragraph 1: as “data” is plural, “The data was…” should be reworded to “The data were…”. Also, the next sentence is missing a possessive apostrophe on participants (e.g. “participants’”).

Discussion 1

* Page 13 paragraph 2: The authors’ added note about how the ownership effect rate observed in their sample differs from that reported in Grisdale et al.’s sample is helpful (although it would be worth specifying if the 95% refers to their neurotypical participants). After highlighting this discrepancy, it would be worth providing some explanation here – why was the rate observed in this study so much lower in comparison with Grisdale et al.?

Experiment 2

* Page 14 paragraph 2: The opening sentence for Experiment 2 is very long and grammatically complex. Recommend breaking up and deleting “- i.e. facilitated by alternative cognitive mechanisms –“ to improve flow.

* Page 17 paragraph 1: remove = from “p = < .001”.

Discussion 2

* Page 19 paragraph 1: I appreciate that the authors now compare categorical percentages of participant groups across experiments. However, the reasoning for similarities/differences are rather speculative and don’t seem particularly convincing. Regarding the difference between corrected hit rates for neurotypical participants in Experiments 1 and 2, it is stated that “This might be partially driven by differences in sample characteristics – in particular, the sample in Experiment 1 was made up entirely of undergraduate students, and furthermore had a higher average AQ (M = 16.98; SD = 6.51; ranging from 2-43) than the neurotypical group in Experiment 2 (M = 14.38; SD = 6.36; ranging from 3-28), both of which might have influenced differences in attention/performance on the task.” Why would we expect undergraduate students to be less likely to show a memory ownership effect than chronologically older non-students? The difference between AQ scores also seems to be an unlikely influence given the absence of statistical relationships between this measure and task performance throughout this study. Also, the autistic participants’ AQ scores in Experiment 2 were much higher and yet a larger % of this group showed an ownership effect than the neurotypical adults in Experiment 1. I recommend that the authors revisit their interpretation here and provide more detailed explanation for their reasoning.

General Discussion

* I think the authors can comfortably claim that the ownership memory effect is reduced in their autistic participants on the basis that corrected hit rate for self-owned items was significantly greater than corrected hit rate for other-owned items in neurotypical participants only. However, the theoretical explanation that this is due to specific deficits associated with self-processing in autism – argued throughout the general discussion – somewhat hinges on p = .06 indicating no population differences for other-owned items. The actual p value may even be lower if reported to 3 dp and the Bayes Factor of 1.72 falls into the category of ‘firm evidence for a difference’ (the same category as 2.31 which is interpreted as a difference in Experiment 1). The groups also did not differ in terms of the proportion of participants showing an ownership effect (the % of autistic participants showing an ownership effect was actually higher than the % of neurotypical participants in Experiment 1). Given the fragility of this evidence, claims for specific deficits in self-processing really ought to be tempered as these data may in fact be indicating that the autistic participants’ average memory accuracy was reduced across the board. I would like to see some acknowledgement and interpretation of this alternative (and arguably more likely) possibility.

* page 21 paragraph 1: As the participant made physical contact with both self- and other-owed items while sorting them, it doesn’t seem relevant to mention contamination when explaining the ownership effect in the context of this study. I also question whether the extended-self hypothesis is relevant here given that participants did not actually own any of the items in this study. Is it really the case that merely sorting a card into a bucket notionally associated with the self is sufficient to transform the card into a marker of one’s personal identity?

* Page 24 paragraph 2: I am still questioning the theoretical interpretation presented here. Awareness of ownership rights is underpinned by sensitivity to cultural rules, so I’m not sure about its relevance. The cited evidence for differences in understanding ownership rights in autism is reported in a paper that tested children with developmental delays. As such, I’m not convinced that this evidence applies to this study’s sample of autistic adults without developmental delays. Likewise, I still have reservations about the authors’ interpretations regarding Theory of Mind (ToM). I don’t disagree with the statement that impairments in ToM might influence awareness of mental states pertaining to owned objects, but how is this relevant to the present study which doesn’t appear to tap into mentalising? Clearer explanation is required in relation to these points, or consider omitting them.

* Page 25 paragraph 2: I appreciated the authors’ more detailed explanation of how they think linguistic cues may have influenced performance across groups. While I think the cited evidence is relevant and useful in terms of situating the present study in relation to wider literature exploring ownership in autism, I am not sure whether the described findings can explain the present study’s results. Again, the mentioned difficulties associated with tracking owner-object relationships using possessive pronouns were reported in autistic children with developmental delay. Is there good reason to expect that they would apply to a sample of autistic adults with high IQ and language skills commensurate with their age? If so, then this needs stronger explanation.

* Page 26 paragraph 2: The authors state that “The current finding that self-referential cognition is diminished in autistic adults is a direct contrast to the patterns reported among autistic children, , where either no group difference or an enhanced effect in the autistic group was found (12,13, Experiment 2). It is hard to reconcile these differences in terms of developmental differences (i.e. that psychological self-awareness is intact in childhood but becomes impaired in adulthood)…” These statements oppose the growing body of evidence for differences in ownership-related cognition in both autistic children and adults.

* Page 27, paragraph 2: It is good to see that the authors have tried to situate their memory tasks in relation to broader ownership effects. I think it would be beneficial to take a further step by reflecting on how the presence and magnitude of between-population differences may (or may not) differ depending on whether stimuli associated with the self and others are truly owned (and why). Also, in light of the authors’ evaluative considerations in this paragraph, do they think it is in fact accurate to consider the memory phenomenon observed in these tasks to be a genuine “ownership effect”? Additionally, the final sentence of this paragraph is a bit unclear (“Therefore, it is unclear whether diminished awareness of self vs. other characteristics would cause impaired ownership effect.”); is this referring to this study’s memory effects specifically, or ownership effects in general? And doesn’t this statement undermine the paper’s theoretical explanation for between-population differences? This paragraph would also benefit from some supporting citations.

General comment: Throughout the manuscript, I would recommend reflecting on wording choices and how they might be perceived by neurodiverse readers, including those who are autistic. Wording like “impoverished self-concept” could potentially be considered offensive.

7. PLOS authors have the option to publish the peer review history of their article (what does this mean?). If published, this will include your full peer review and any attached files.

Reviewer #1: **Yes: **Calum Hartley

---

## [Author Response · Author response to Decision Letter 1]

2 Oct 2023

* Page 4 paragraph 2: When describing Grisdale et al. (2014) it is stated that neurotypical participants showed a 9% advantage in memory for self-owned items over other-owned items. When describing the results of the autistic participants, it would be helpful to also include a % for accurate comparison.

This has been added. 

* Page 4 paragraph 2: New text at the end of para 2 seems to be presented in a smaller font.

This has been corrected. 

* Page 5 paragraph 2: Possible typo “Firstly, neurotypical adults studies…” perhaps reword to “Firstly, studies of neurotypical adults”?

This has been corrected. 

* Page 5 paragraph 2: With the added text, sentence 2 is extremely long. Recommend breaking up by starting a new sentence from “Additionally, in the perceptual domain…”

This has been corrected. 

* Page 11, footnote: For readers who are unfamiliar with d-prime, it might be worth providing a brief definition and explanation for its relevance to this study.

We have added a definition to the footnote on page 10-11. 

* Page 12 paragraph 1: as “data” is plural, “The data was…” should be reworded to “The data were…”. Also, the next sentence is missing a possessive apostrophe on participants (e.g. “participants’”).

This has been corrected.

* Page 13 paragraph 2: The authors’ added note about how the ownership effect rate observed in their sample differs from that reported in Grisdale et al.’s sample is helpful (although it would be worth specifying if the 95% refers to their neurotypical participants). After highlighting this discrepancy, it would be worth providing some explanation here – why was the rate observed in this study so much lower in comparison with Grisdale et al.?

We have added an explanation to highlight likely causes for between-study differences in the size of the effect on page 12: “This difference is likely explained by subtle differences between studies in task instructions, the larger sample size in our study compared to Grisdale et al. (N=100 vs. 40), and sample demographics.”

* Page 14 paragraph 2: The opening sentence for Experiment 2 is very long and grammatically complex. Recommend breaking up and deleting “- i.e. facilitated by alternative cognitive mechanisms –“ to improve flow.

This has been corrected.

* Page 17 paragraph 1: remove = from “p = < .001”.

This has been removed.

* Page 19 paragraph 1: I appreciate that the authors now compare categorical percentages of participant groups across experiments. However, the reasoning for similarities/differences are rather speculative and don’t seem particularly convincing. Regarding the difference between corrected hit rates for neurotypical participants in Experiments 1 and 2, it is stated that “This might be partially driven by differences in sample characteristics – in particular, the sample in Experiment 1 was made up entirely of undergraduate students, and furthermore had a higher average AQ (M = 16.98; SD = 6.51; ranging from 2-43) than the neurotypical group in Experiment 2 (M = 14.38; SD = 6.36; ranging from 3-28), both of which might have influenced differences in attention/performance on the task.” Why would we expect undergraduate students to be less likely to show a memory ownership effect than chronologically older non-students? The difference between AQ scores also seems to be an unlikely influence given the absence of statistical relationships between this measure and task performance throughout this study. Also, the autistic participants’ AQ scores in Experiment 2 were much higher and yet a larger % of this group showed an ownership effect than the neurotypical adults in Experiment 1. I recommend that the authors revisit their interpretation here and provide more detailed explanation for their reasoning.

Thank you for pointing out this inconsistency in our explanation. We have revised this paragraph as follows: “These differences might be influenced by differences in sample characteristics. In particular, the neurotypical sample in Experiment 1 was made up entirely of undergraduate students who completed the task for course credit, while participants in Experiment 2 were recruited from the Autism Research Kent participant database (made up of members of the local community who are interested in research) and were paid for their time. These different sample contexts might have influenced participants’ attention/performance on the task due to differing levels of motivation (Hess, 2014; Freund, 2008).”

* I think the authors can comfortably claim that the ownership memory effect is reduced in their autistic participants on the basis that corrected hit rate for self-owned items was significantly greater than corrected hit rate for other-owned items in neurotypical participants only. However, the theoretical explanation that this is due to specific deficits associated with self-processing in autism – argued throughout the general discussion – somewhat hinges on p = .06 indicating no population differences for other-owned items. The actual p value may even be lower if reported to 3 dp and the Bayes Factor of 1.72 falls into the category of ‘firm evidence for a difference’ (the same category as 2.31 which is interpreted as a difference in Experiment 1). The groups also did not differ in terms of the proportion of participants showing an ownership effect (the % of autistic participants showing an ownership effect was actually higher than the % of neurotypical participants in Experiment 1). Given the fragility of this evidence, claims for specific deficits in self-processing really ought to be tempered as these data may in fact be indicating that the autistic participants’ average memory accuracy was reduced across the board. I would like to see some acknowledgement and interpretation of this alternative (and arguably more likely) possibility.

Thank you for raising this important distinction. The group difference in memory for other-owned items has a p-value of 0.063 (we now report this to three decimal places in the paper for clarity). We have also expanded the discussion of Experiment 2 to highlight the subtleties in effects across different measures, and propose the possibility that a more general memory accuracy impairment might exist in the autistic group. 

* page 21 paragraph 1: As the participant made physical contact with both self- and other-owned items while sorting them, it doesn’t seem relevant to mention contamination when explaining the ownership effect in the context of this study. I also question whether the extended-self hypothesis is relevant here given that participants did not actually own any of the items in this study. Is it really the case that merely sorting a card into a bucket notionally associated with the self is sufficient to transform the card into a marker of one’s personal identity?

We have removed this section from the manuscript. 

* Page 24 paragraph 2: I am still questioning the theoretical interpretation presented here. Awareness of ownership rights is underpinned by sensitivity to cultural rules, so I’m not sure about its relevance. The cited evidence for differences in understanding ownership rights in autism is reported in a paper that tested children with developmental delays. As such, I’m not convinced that this evidence applies to this study’s sample of autistic adults without developmental delays. Likewise, I still have reservations about the authors’ interpretations regarding Theory of Mind (ToM). I don’t disagree with the statement that impairments in ToM might influence awareness of mental states pertaining to owned objects, but how is this relevant to the present study which doesn’t appear to tap into mentalising? Clearer explanation is required in relation to these points, or consider omitting them.

We have cut this paragraph. 

* Page 25 paragraph 2: I appreciated the authors’ more detailed explanation of how they think linguistic cues may have influenced performance across groups. While I think the cited evidence is relevant and useful in terms of situating the present study in relation to wider literature exploring ownership in autism, I am not sure whether the described findings can explain the present study’s results. Again, the mentioned difficulties associated with tracking owner-object relationships using possessive pronouns were reported in autistic children with developmental delay. Is there good reason to expect that they would apply to a sample of autistic adults with high IQ and language skills commensurate with their age? If so, then this needs stronger explanation.

We have edited these paragraphs to highlight the key arguments relating to linguistic cues and remove those that do not provide a strong enough explanation. 

* Page 26 paragraph 2: The authors state that “The current finding that self-referential cognition is diminished in autistic adults is a direct contrast to the patterns reported among autistic children, where either no group difference or an enhanced effect in the autistic group was found (12,13, Experiment 2). It is hard to reconcile these differences in terms of developmental differences (i.e. that psychological self-awareness is intact in childhood but becomes impaired in adulthood)…” These statements oppose the growing body of evidence for differences in ownership-related cognition in both autistic children and adults.

We have now qualified this discussion by adding: “Nevertheless, the results from the current study are in keeping with the growing body of evidence showing differences in autistic children/adults ownership-related cognition in non-memory-related tasks (10,46,48,49)” at the end of this paragraph – page 27. 

* Page 27, paragraph 2: It is good to see that the authors have tried to situate their memory tasks in relation to broader ownership effects. I think it would be beneficial to take a further step by reflecting on how the presence and magnitude of between-population differences may (or may not) differ depending on whether stimuli associated with the self and others are truly owned (and why). 

We have raised this discussion point on page 26.

Also, in light of the authors’ evaluative considerations in this paragraph, do they think it is in fact accurate to consider the memory phenomenon observed in these tasks to be a genuine “ownership effect”?

We have amended this point to clarify that it might reflect something qualitatively different than an ownership effect that is developed organically when items are genuinely owned by the self.

Additionally, the final sentence of this paragraph is a bit unclear (“Therefore, it is unclear whether diminished awareness of self vs. other characteristics would cause impaired ownership effect.”); is this referring to this study’s memory effects specifically, or ownership effects in general? And doesn’t this statement undermine the paper’s theoretical explanation for between-population differences? This paragraph would also benefit from some supporting citations.

This line has been deleted, and references have been added. 

General comment: Throughout the manuscript, I would recommend reflecting on wording choices and how they might be perceived by neurodiverse readers, including those who are autistic. Wording like “impoverished self-concept” could potentially be considered offensive.

We have checked terminology throughout the paper to avoid potentially stigmatising language, replacing “impoverished” with “diminished”, and removed all mention of “deficit” and replaced it with “impairment” or “atypicality”.

---

## [Editor Report · Decision Letter 2]

23 Oct 2023

Revisiting the Ownership Effect in Adults with and without Autism

PONE-D-23-08426R2

Dear Dr. Ferguson,

We’re pleased to inform you that your manuscript has been judged scientifically suitable for publication and will be formally accepted for publication once it meets all outstanding technical requirements.

Kind regards,

Johannes Hönekopp

Academic Editor

PLOS ONE
---

## [Editor Report · Acceptance letter]

3 Nov 2023

PONE-D-23-08426R2 

Revisiting the Ownership Effect in Adults with and without Autism 

Dear Dr. Ferguson:

I'm pleased to inform you that your manuscript has been deemed suitable for publication in PLOS ONE. Congratulations! Your manuscript is now with our production department. 

Kind regards, 

on behalf of

Dr. Johannes Hönekopp 

Academic Editor

PLOS ONE